# Excitatory to inhibitory synaptic ratios are unchanged at presymptomatic stages in multiple models of ALS

Calum Bonthron[1], Sarah Burley[1,2], Matthew J. Broadhead[1], Vanya Metodieva[2,3], Seth G. N. Grant[4,5], Siddharthan Chandran[4,5,6,7], Gareth B. Miles[1]*

**1** School of Psychology and Neuroscience, University of St Andrews, St Andrews, United Kingdom, **2** School of Biology, University of St Andrews, St Andrews, United Kingdom, **3** Centre of Biophotonics, University of St Andrews, St Andrews, United Kingdom, **4** Centre for Clinical Brain Sciences, University of Edinburgh, Edinburgh, United Kingdom, **5** Simons Initiative for the Developing Brain (SIDB), Centre for Discovery Brain Sciences, University of Edinburgh, Edinburgh, United Kingdom, **6** UK Dementia Research Institute, Edinburgh Medical School, University of Edinburgh, Edinburgh, United Kingdom, **7** Patrick Wild Centre, University of Edinburgh, Edinburgh, United Kingdom

* gbm4@st-andrews.ac.uk

**Data Availability Statement:** The data underpinning this publication can be accessed via the University of St Andrews Research Portal

## Abstract

Hyperexcitability of motor neurons and spinal cord motor circuitry has been widely reported in the early stages of Amyotrophic Lateral Sclerosis (ALS). Changes in the relative amount of excitatory to inhibitory inputs onto a neuron (E:I synaptic ratio), possibly through a developmental shift in synapse formation in favour of excitatory transmission, could underlie pathological hyperexcitability. Given that astrocytes play a major role in early synaptogenesis and are implicated in ALS pathogenesis, their potential contribution to disease mechanisms involving synaptic imbalances and subsequent hyperexcitability is also of great interest. In order to assess E:I ratios in ALS, we utilised a novel primary spinal neuron / astrocyte co-culture system, derived from neonatal mice, in which synapses are formed *in vitro*. Using multiple ALS mouse models we found that no combination of astrocyte or neuron genotype produced alterations in E:I synaptic ratios assessed using pre- and post-synaptic anatomical markers. Similarly, we observed that ephrin-B1, a major contact-dependent astrocytic synaptogenic protein, was not differentially expressed by ALS primary astrocytes. Further to this, analysis of E:I ratios across the entire grey matter of the lumbar spinal cord in young (post-natal day 16–19) ALS mice revealed no differences versus controls. Finally, analysis in co-cultures of human iPSC-derived motor neurons and astrocytes harbouring the pathogenic C9orf72 hexanucleotide repeat expansion showed no evidence of a bias toward excitatory versus inhibitory synapse formation. We therefore conclude, utilising multiple ALS models, that we do not observe significant changes in the relative abundance of excitatory versus inhibitory synapses as would be expected if imbalances in synaptic inputs contribute to early hyperexcitability.

(https://doi.org/10.17630/79bcc046-8043-4c37-ab7a-21326f0bf1ad).

**Funding:** Motor Neurone Disease (MND) Association UK (authors: M.J.B., G.B.M., S.C.; grant number: Miles/Apr18/863-791; URL: https://www.mndassociation.org/), the SPRINT MND/MS PhD Programme (authors: C.B.; URL: https://www.edinburghneuroscience.ed.ac.uk/edneurophd/sprint-mndms-phd-programme), Wellcome Trust (authors: S.G.N.G; grant number: Technology Development Grant 202932; URL:https://wellcome.org/), the European Union Seventh Framework Programme (authors: S.G.N.G.; grant numbers: HEALTH- F2-2009-241498 and 720270; URL: https://ec.europa.eu/commission/presscorner/detail/hu/MEMO_16_146) and the European Research Council (ERC) under the European Union's Horizon 2020 research and innovation programe (authors: S.G.N.G.; grant number: 695568, URL: https://erc.europa.eu/homepage).

**Competing interests:** The authors have declared that no competing interests exist.

## Introduction

Early hyperexcitability of motor neurons (MNs), thought to be due in part to altered intrinsic properties, is an often-cited mechanism which may contribute to excitotoxic cell death in ALS, and has been reported in a variety of *in vitro* and *ex vivo* studies [1–6]. Excessive excitability of wider spinal interneuron populations has been reported too [3, 7, 8], indicating that this phenomenon may not be exclusive to MNs, but rather observed throughout the pre-MN circuitry. As the excitability of a neuron is partially dictated by the balance of excitatory to inhibitory synaptic inputs, an imbalance of this ratio has the potential to elicit pathogenic hyperexcitability. Insufficient inhibitory input, or excessive excitatory input, either onto the MNs themselves or within pre-MN networks, may lead to ALS-related excitotoxic neuronal death [9]. Anatomical work has long implicated synaptic pathology in ALS, with it being apparent in early post-mortem studies [10–13]. Indeed, studies in various disease models have demonstrated a reduction in the density of inhibitory innervation onto MNs [14–18].

Whilst it has been posited that there is a shift in E:I ratios, likely due to inhibitory loss during disease pathogenesis [9], there is an alternative hypothesis for the basis of this change. Instead of a later shift towards excitation due to synaptic loss, there may in fact be a developmental alteration in ALS that biases the system towards excitation from early life [19]. Although neurodegenerative conditions are not classically thought of as such, ALS could be considered a developmental disorder, as is proposed for other neurological conditions such as schizophrenia and autism spectrum disorders [20]. Such an early developmental bias in E:I ratios throughout the spinal circuitry could explain observations of spinal circuit hyperexcitability.

Pathological changes in synaptogenesis could be driven by astrocytes. Astrocyte-mediated synaptogenesis occurs postnatally, occurring maximally in weeks 2 and 3 in the brain, and showing a similar pattern in the spinal cord which appears to peak at the end of week 2 / start of week 3 before declining [21, 22]. Astrocytes promote synapse formation in a complex manner, utilising both secreted factors such as thrombospondins and hevin, as well as contact-dependent factors such as γ-protocadherins and ephrin-B1 [21, 23, 24]. Although studies of astrocyte-mediated synaptogenesis typically focus on excitatory synapse formation, astrocytes are also directly involved in inhibitory synaptogenesis [25–27]. It is therefore clear that astrocytes have complex signalling mechanisms which allow them to modulate both excitatory and inhibitory synapse formation that could contribute to early hyperexcitability in ALS.

To look for evidence of early changes in E:I synaptic ratios, dependent or independent of astrocytic pathology, we quantified excitatory and inhibitory synapse numbers from multiple ALS models using high-resolution fluorescence microscopy. These included novel primary co-cultures of spinal neurons and astrocytes from postnatal SOD1$^{G93A}$ and C9orf72 (C9BAC500) mutant mice, spinal cord tissue from SOD1$^{G93A}$ animals immediately after the major peak of postnatal synaptogenesis, and human induced pluripotent stem cell (iPSC)-derived co-cultures of MNs and astrocytes harbouring C9orf72 hexanucleotide expansions.

## Materials and methods

### Animals and Ethics

All procedures performed on animals were done in accordance with the UK Animals (Scientific Procedures) Act 1986 and approved by the University of St Andrews Animal Welfare and Ethics Committee. Both ALS mouse lines were provided by Dr. Richard Mead (University of Sheffield): B6SJL-TgN(SOD1-G93A)1Gur/J (SOD1$^{G93A}$) and FVB/NJ-Tg(C9ORF72)500Lpwr/

J (C9BAC500). These were both bred with PSD95-eGFP$^{+/+}$ animals (provided by Prof. Seth Grant) to produce offspring that all expressed PSD95-eGFP$^{+/-}$. As C9BAC500 animals were maintained on an FVB background strain, all C9BAC500 x PSD95-eGFP animals used were first generation FVB/C57bl/6J crosses.

## Harvesting of ALS x PSD95-eGFP spinal cord tissue

P16-19 SOD1$^{G93A}$ x PSD95-eGFP progeny were injected with euthanal (Merial Animal Health) before transcardial perfusion with 1xPBS (Gibco) then 4% paraformaldehyde (PFA) (Alfa Aesar). Spinal cords were dissected out and placed in 4% PFA for a further 4 hours. They were then washed in PBS, before being left in 30% sucrose solution overnight at 4˚C. Finally, they were incubated in an equal mixture of sucrose solution / OCT (optical cutting temperature) embedding compound (Scigen) for 2 hours. Lumbar spinal cords were then trimmed and embedded in OCT at -80˚C. They were sliced at approximately 20µm thickness using a Leica CM1860 cyrostat and mounted onto Superfrost Gold Plus glass slides (VWR).

## Image analysis and statistics

Quantification of synapses was performed in Fiji is just ImageJ (FIJI) [28]. For all *in vitro* work (primary co-culture and iPSC-derived MNs), images were processed using background subtraction and gaussian smoothing to increase clarity of synaptic puncta. For detection of synapsin, Moments-based thresholding was utilised, while for PSD95 and gephyrin Otsu-based thresholding was used. Typically, a degree of manual adjustment was required. The experimenter was blinded to conditions (genotype) during imaging and analysis to avoid unconscious bias. Minimum and maximum size thresholds were set to avoid detection of both very dim structures, as well as large supra-synaptic sized objects that were likely lipofuscin aggregates or other debris. Thresholding produced binarised images, with the overlap of pre- and postsynaptic objects (by at least 1 pixel) being interpreted as a synapse.

Immunohistochemical work in mouse tissue was done similarly, including image processing steps and blinding. Delineation of Rexed's laminae was completed using a standard mouse spinal cord anatomical atlas as reference [29]. For clarity, lamina 8 contains a heterogenous population of ventral neurons including the medial MN pool, whilst lamina 9 contains the lateral MN pool. Thresholding was completed on a lamina-by-lamina basis in these data sets in order to accurately detect all structures within each region without variability of expression between laminae causing dimmer puncta to be missed.

Data processing was performed in Microsoft Excel. Statistical analysis and graph preparation were performed in Prism 10 (Graphpad). Data was assessed for normality using the Shapiro-Wilk test, and when applicable for two-way ANOVAs, Geisser-Greenhouse corrected values were calculated as sphericity was not assumed. Two-way ANOVAs were conducted for all normalised synapse counts, tissue E:I ratio quantification, synaptic mapping and ephrin-B1 quantification. Multiple comparisons testing was done using either Šdák or Tukey corrections as recommended depending on the number of comparisons conducted. Between-subjects, one-way ANOVAs were used for primary culture E:I ratios and isolated excitatory / inhibitory synapse comparisons. If parametric assumptions were not met, Kruskal-Wallis tests were used instead. Unpaired, two-tailed t-tests were used to assess individual differences in excitatory and inhibitory synapse counts in iPSC-derived cultures. Statistical significance is denoted as follows: * = <0.05, ** = <0.01, *** = <0.001, **** = <0.0001.

## Immunocytochemistry—Primary co-cultures + iPSC-Derived MN / astrocyte cultures

Coverslips were washed in Dulbecco's Phosphate-Buffered Saline (DPBS) (Gibco), and fixed in 4% PFA for 25 mins, before 3 further DPBS washes. Cells were then permeabilised in DPBS w. 0.1% Triton X for 10 minutes. Next, DPBS w. 5% Bovine Serum Albumin (BSA) / 0.1% Triton X was added, and the cells were blocked for 1 hour. Primary antibodies were diluted in this solution (see Table 1) and left to incubate overnight at 4˚C. 3 5-minute washes were then conducted in DPBS w. 0.1% Triton X, before dilution of secondary antibodies (see Table 1) in DPBS w. 5% BSA / 0.1% Triton X. Coverslips were incubated in this solution for 1 hour at room temperature, before being washed 3 times in DPBS. Finally, 1:5000 DAPI solution was added (Stock = 5mg/ml, Sigma Aldrich) for 10 minutes, before a final DPBS wash. Coverslips were mounted face down onto microscopy slides with Vectashield Vibrance (Vector Labs) mounting media.

## Immunohistochemistry—Spinal tissue

Slides were placed in an incubator at 37˚C for 30 min to aid adherence of tissue to the glass slides and reduce loss during washing steps. These were then washed 3 times in PBS, before application of blocking solution (PBS w. 3% BSA and 0.25% Triton X) for 2 hours. Primary antibodies (see Table 1) were diluted in PBS w. 1.5% BSA / 0.125% Triton X) and added for 2 nights at 4˚C. Primary antibody solution was then removed, slides washed 5 times in PBS, then secondary ABs (see Table 1) were added diluted in PBS w. 0.1% Triton X. Incubation occurred over 2 hours at room temperature, before a further 5 PBS washes. Slides were then washed in dH$_2$O and mounted in Mowiol with 13mm coverslips.

**Table 1. Information on utilised primary and secondary antibodies.**

| Primary AB | Species | Company / Product Code | Tissue Conc. | *In Vitro* Conc. | Secondary AB | Species | Company | Tissue Conc. | *In Vitro* Conc. |
|---|---|---|---|---|---|---|---|---|---|
| Ephrin-B1 | Goat | Bio-Techne / AF473 | N/A | 1:500 | anti-Goat Alexa Fluor 488 | Donkey | Invitrogen / A-11055 | N/A | 1:1000 |
| GAD65 | Goat | Abcam / ab112007 | N/A | 1:500 | anti-Goat Alexa Fluor 488 | Donkey | Invitrogen / A-11055 | N/A | 1:1000 |
| Gephyrin | Mouse | Synaptic Systems / 147021 | 1:500 | 1:500 | anti-Mouse Alexa Fluor 555 | Donkey | Invitrogen / A-31570 | 1:500 | 1:1000 |
| GFAP | Chicken | Aves Labs / GFAP | N/A | 1:500 | anti-Chicken Alexa Fluor Plus 647 | Goat | Invitrogen– A32933 | N/A | 1:1000 |
| Glutamine Synthetase | Mouse | ProteinTech / 66323-1-Ig | N/A | 1:500 | anti-Mouse Alexa Fluor 555 | Donkey | Invitrogen / A-31570 | N/A | 1:1000 |
| PSD95 | Guinea Pig | Synaptic Systems / 124014 | N/A | 1:500 | anti-Guinea Pig Alexa Fluor 647 | Goat | Abcam / ab150187 | N/A | 1:1000 |
| PSD95 | Mouse | Abcam / ab2723 | N/A | 1:500 | anti-Mouse Alexa Fluor 555 | Donkey | Invitrogen / A-31570 | N/A | 1:1000 |
| Synapsin | Rabbit | Cell Signalling / 52978 | 1:500 | 1:500 | anti-Rabbit Alexa Fluor Plus 647 | Donkey | Invitrogen / A32795 | 1:500 | 1:1000 |
| | | | | | anti-Rabbit Alexa Fluor Plus 488 | Goat | Invitrogen / A32731 | N/A | 1:1000 |
| VGAT | Rabbit | Proteintech / 14471-1-AP | N/A | 1:500 | anti-Rabbit Alexa Fluor Plus 647 | Donkey | Invitrogen / A32795 | N/A | 1:1000 |

## iPSC-Derived MN / astrocyte culture

Patient-derived iPSCs were produced from dermal fibroblasts obtained under Ethical / Institutional Review Board approval at the University of Edinburgh. The reprogramming of fibroblasts into iPSCs, as well as generation of the isogenic control line, were performed as described previously [30]. iPSC-derived cultures were produced using a slightly modified protocol described by Selvaraj and colleagues [30]. At the beginning of MN differentiation (DIV 0), TrypLE (Gibco) was used in place of Accutase during iPSC dissociation. During MN plating (DIV 16), a 1:30 Matrigel (Corning) solution was used instead of the previously used poly-ornithine, laminin, fibronectin and matrigel combination. Treatment of plated MNs with Uridine / 5-fluoro-2'-deoxyuridine (U/FDU) was emitted to enable astrocyte proliferation and creation of a mixed culture. Additionally, cells were switched from 'NB media' to BrainPhys (Stemcell Technologies) at 2 weeks post-plating (DIV30) with Neurocult SM1 (Stemcell Technologies), N2 supplement (Gibco), in addition to previously utilised maturation factors.

## Primary co-culture of postnatal neonatal mice

**Astrocytes.** P2-4 animals were cervically dislocated, decapitated and their lumbar spinal cords were dissected free-floating. Cords were stripped of their meninges, diced and enzymatically dissociated. The enzyme from the 'NeuroCult Enzyme Dissociation Kit for Adult CNS Tissue' (Stemcell Technologies) was toxic to neural populations. The tissue was then washed and agitated multiple times to produce a single cell solution that could be plated in T75 flasks, grown in astrocyte media consisting of DMEM + Glutamax (Gibco), Fetal Bovine Serum (FBS) and Antibiotic-Antimycotic (Gibco). 8–10 days later when confluent, astrocytes were replated onto coverslips.

**Co-Cultures.** P2-3 animals were killed and dissected as described for astrocytes. Alternative enzymatic dissociation utilising a papain latex solution was conducted to preserve neuronal populations, again followed by washing and agitation steps. Suspensions were centrifuged, before counting and plating directly onto the desired genotype of astrocyte with antimitotic agent U/FDU (Sigma-Aldrich) to eliminate the majority of astrocytic populations in the neuronal monolayer, mixed into Neurobasal A supplemented with B27 (Gibco), Glutamax (Gibco) and Antiobiotic-Antimycotic. Cultures received another full feed with U/FDU 2–3 days later, before switching to negative media consisting of BrainPhys supplemented with Neurocult SM1 and Antibiotic-Antimycotic.

## Whole-cell patch-clamp recording

Coverslips were placed in a recording chamber, perfused with recirculating oxygenated artificial cerebral spinal fluid (0.127M NaCl, 0.003M KCl, 0.00125M $NaH_2PO_4$, 0.001M $MgCl_2$, 0.002M $CaCl_2$, 0.026M $NaHCO_3$ and 0.01M Glucose, equilibrated with 95% $O_2$ and 5% $CO_2$, pH 7.45, osmolarity approx. 310mOsm) at room temperature (approx 20˚C). Patch electrodes were pulled using a Sutter P-97 horizontal puller (Sutter Instrument Company) from borosilicate glass capillaries (World Precision Instruments), with electrodes having a resistance range of approximately 5-8MΩ. Electrodes were filled with internal recording solution (0.14M $KMeSO_4$, 0.01M NaCl, 0.001M $CaCl_2$, 0.01M HEPES, 0.001M EGTA, 0.003M Mg-ATP and 0.0004M GTP, pH 7.2–7.3 adjusted with KOH, osmolarity 300mOsm adjusted with sucrose). Cells were visualised using an Olympus upright BX51WI microscope equipped with a x40 submersion lens. Signals were amplified and filtered using a MultiClamp 700B amplifier (Axon Instruments) and acquired using a Digitdata 1440A analogue-to-digital board with pClamp (Axon Instruments). Clampfit 11 was used to visualise traces and aid in manual analysis.

## Results

### Astrocytes harbouring ALS mutations do not influence E:I ratios in primary mouse co-cultures

To assess the effects of astrocytes on E:I ratios during synaptic formation in ALS, we developed a neuron-astrocyte co-culture system in which new synapses are formed in the presence of astrocytes of different genotypes. First, we separately cultured astrocytes from postnatal mice. Then astrocytes and neurons were combined in assigned genotype combinations. We developed techniques to obtain cells from postnatal animals because of advantages known from postnatal protocols of other CNS regions, including a reduction of animals sacrificed according to the 3Rs, and the ability to genotype transgenic animals before platedown [31]. We utilised two different mouse models of ALS, SOD1^G93A [32] and C9BAC500 [33] animals. Both models were bred with mice expressing GFP fused to the postsynaptic density protein PSD95 [34, 35]. This enabled visualisation of excitatory postsynaptic structures, and when combined with immunolabelling of an equivalent inhibitory postsynaptic marker (gephyrin) [36, 37] and a ubiquitously expressed presynaptic marker (synapsin) [38], we could calculate E:I ratios in our co-cultures.

We began by producing glial monocultures and validated their degree of astrocytic enrichment using multiple markers (see S1 Fig). Typically, 2–3 weeks later, neurons +ve or -ve for the SOD1^G93A or C9BAC500 mutations were plated on top of astrocytes and cultured until DIV 21. This time point was chosen as this is when neural monocultures showed evidence of reliable spontaneous synaptic activity and robust PSD95 expression (see S2 Fig). Neurons in our co-cultures showed punctate expression of synapsin, PSD95-eGFP and gephyrin (see Fig 1A). Concomitant PSD95 antibody staining also confirmed PSD95-eGFP puncta were expressed as expected (see S3 Fig). Importantly, we saw significant juxtaposition of pre- and post-synaptic markers, which allowed us to quantify the ratio of excitatory (synapsin-PSD95) to inhibitory (synapsin-gephyrin) synapses. For brevity, +N and +A refer to neurons and astrocytes, respectively, which are positive for the SOD1^G93A or C9BAC500 mutation. -N and -A refers to these cell types generated from negative littermates that do not express ALS mutations. This produced 4 conditions in which to investigate non-cell autonomous effects on E:I ratios (-N/-A, -N/+A, +N/-A, +N/+A) (see Fig 1B).

We first investigated the potential effects of the SOD1^G93A mutation on E:I ratios, as this provides the clearest and simplest metric for assessing the balance between excitatory and inhibitory synapses in our cultures. E:I ratios were calculated by quantifying the number of excitatory and inhibitory synapses in each image and calculating a ratio (no. excitatory synapses / no. inhibitory synapses) for each field of view. Analysis of excitatory and inhibitory synapses revealed no significant differences in the E:I synaptic ratio between any of the 4 conditions ($F(3,12) = 0.9431$, P = 0.45), whether the SOD1^G93A mutation was expressed in neurons, astrocytes or both. In order to verify that the static E:I was not masking a scaled increase / decrease of either synapse type, we examined counts of excitatory and inhibitory synapses separately. No differences in excitatory or inhibitory synapse counts (normalised to area; density per. $100\mu m^2$) were observed between genotype conditions (excitatory: $F(3,12) = 0.7563$, P = 0.76; inhibitory: $F(3,12) = 0.1308$, P = 0.94). When analysed together, our analyses revealed that all cultures contained a greater number of inhibitory synapses than excitatory synapses ($F(1,24) = 8.708$, P = 0.0070), with mean E:I ratios ranging from 0.62–0.74 across conditions (see Fig 1B and 1C). Reflective of the lack of observed difference in E:I, the degree of inhibitory bias was not dependent on the genotype ($F(3,24) = 0.01942$, P = 0.99).

In order to account for potential variability in the number of neurons within the different images analysed, synapse numbers were also normalised to DAPI-stained nuclei strictly in the

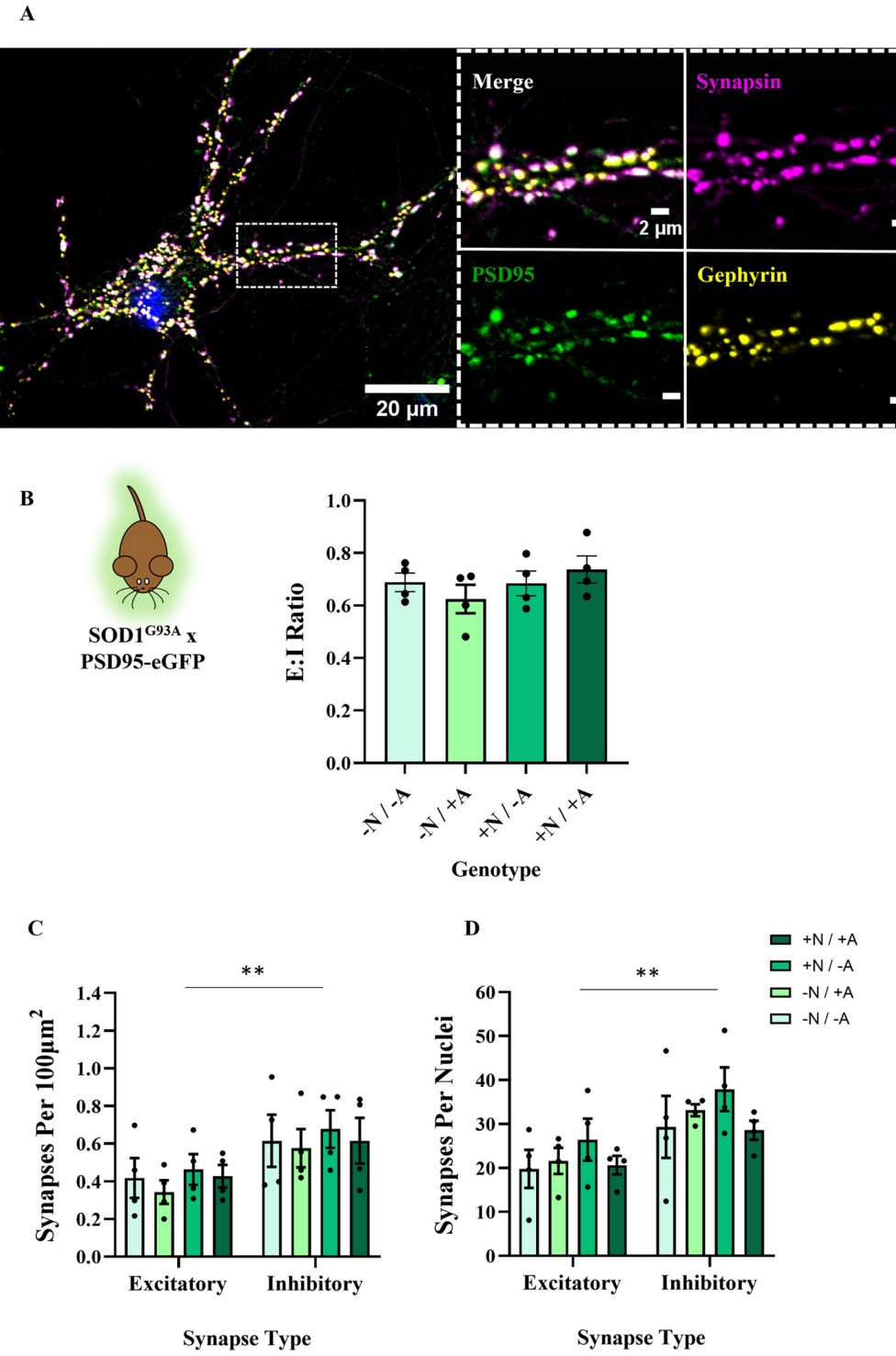

**Fig 1. The presence of the SOD1^G93A mutation in astrocytes or neurons has no impact on E:I ratios in primary co-cultures. A**) Example image of DIV 21 neuron in a SOD1$^{G93A}$ x PSD95-eGFP co-culture expressing synapsin, PSD95-eGFP and gephyrin. When zoomed, we see clear co-localisation of presynaptic and postsynaptic markers. **B**) Bar chart showing E:I ratios in co-cultures of 4 different combinations. + /—refers to the presence of the SOD1$^{G93A}$ mutation, N / A refers to the cell type referenced, either neurons or astrocytes, respectively (N = 4 co-culture platedowns). **C**) Bar chart showing numbers of excitatory and inhibitory synapses normalised to area (N = 4 co-culture platedowns). **D**) Bar chart showing numbers of excitatory and inhibitory synapses normalised to nuclei count (N = 4 co-culture platedowns).

focal plane of synaptic protein expression. Consistent with data normalised to area, there was no difference in DAPI-normalised excitatory or inhibitory synapse counts between genotype conditions (excitatory: $F(3,12) = 0.6471$, P = 0.60; inhibitory: $F(3,12) = 0.9013$, P = 0.47). Again, we observed higher numbers of inhibitory versus excitatory synapses in these cultures when normalised to DAPI nuclei ($F(1,24) = 12.09$, P = 0.0020) (see Fig 1D), and the degree of this inhibitory bias was not dependent on genotype condition ($F(3,24) = 0.08735$, P = 0.97).

We next investigated whether the C9orf72 hexanucleotide repeat, present in C9BAC500 mice [33] affected synaptic counts or E:I ratios. We found no difference in the E:I ratios between our 4 genotype conditions ($F(3,21) = 1.667$, P = 0.20) (see Fig 2A and 2B). There was no difference in excitatory or inhibitory synapse numbers between conditions, whether data were normalised by area (excitatory: $F(3,21) = 0.9561$, P = 0.43; inhibitory: $F(3,21) = 1.462$, P = 0.25) or by DAPI count (excitatory: $X^2(3) = 4.412$, P = 0.22; inhibitory: $X^2(3) = 5.127$, P = 0.16). Once more, there was a bias towards inhibitory synapse formation in these cultures (area: $F(1,42) = 13.05$, P = 0.0008; DAPI: $F(1,42) = 9.532$, P = 0.0036), with an E:I range of 0.61–0.78 (see Fig 2C and 2D). Genotype did not affect the degree of this inhibitory bias (area: $F(3,42) = 0.3959$, P = 0.76; DAPI: $F(3,42) = 0.3139$, P = 0.82). When compared to our SOD1$^{G93A}$ cultures, there was no difference in E:I ratios depending on ALS mouse model used ($F(1,33) = 0.0006632$, P = 0.94).

Our experiments on spinal neuron and astrocyte co-cultures generated from both SOD1$^{G93A}$ and C9BAC500 animals therefore demonstrate that the presence of an ALS mutation does not alter the E:I synaptic ratio, or the number of excitatory or inhibitory synapses that develop in this cell culture model.

## Astrocytic ephrin-B1 expression is not affected by ALS mutations

Ephrin-B1 expression in astrocytes has been demonstrated to modulate the E:I balance during hippocampal synaptogenesis [24, 39, 40]. This ligand interacts with ephrin receptors on neurons in a contact-dependent manner enabling cell-cell communication [21]. Expression of astrocytic ephrin-B1 directly influences both excitatory and inhibitory synapse numbers, whereby reduced expression biases the system towards an increased E:I ratio [24].

Although we did not observe any changes in excitatory or inhibitory synapse numbers in the cultured spinal neurons from ALS mouse models, it was hypothesised that cultured astrocytes from ALS mouse models may show changes in ephrin-B1 expression that could contribute to ALS synaptopathy at later stages in development. We therefore immunolabelled ephrin-B1 in cultured astrocytes prepared from SOD1$^{G93A}$ mice, C9BAC500 mice and respective control littermates (see Fig 3A). No difference in ephrin-B1 expression was observed between control and disease astrocytes across 2–6 weeks *in vitro*, whether measuring the number of ephrin-B1 particles (SOD1$^{G93A}$: $F(1,4) = 0.02301$, P = 0.89; C9BAC500: $F(1,6) = 1.909$, P = 0.22), the summated (integrated) intensity of the particles (SOD1$^{G93A}$: $F(1,4) = 1.928$, P = 0.24; C9BAC500: $F(1,6) = 0.4855$, P = 0.51) or the overall intensity of ephrin-B1 expression from whole images (field intensity) (SOD1$^{G93A}$: $F(1,4) = 1.567$, P = 0.28; C9BAC500: $F(1,6) = 0.3940$, P = 0.55) (see Fig 3B and 3C). This lack of change in ephrin-B1 in SOD1$^{G93A}$ or C9BAC500 astrocyte cultures provides further evidence that postnatal ALS astrocytes do not cause non-cell autonomous synaptogenic effects on E:I ratios in disease.

## E:I ratios are unchanged in the young SOD1$^{G93A}$ mouse spinal cord

As we observed no changes in E:I ratios when new synapses were being formed using our co-culture model, we next wanted to verify this was also the case in the lumbar spinal cord of young SOD1$^{G93A}$ mice. Postnatal development of the spinal circuitry is associated with major

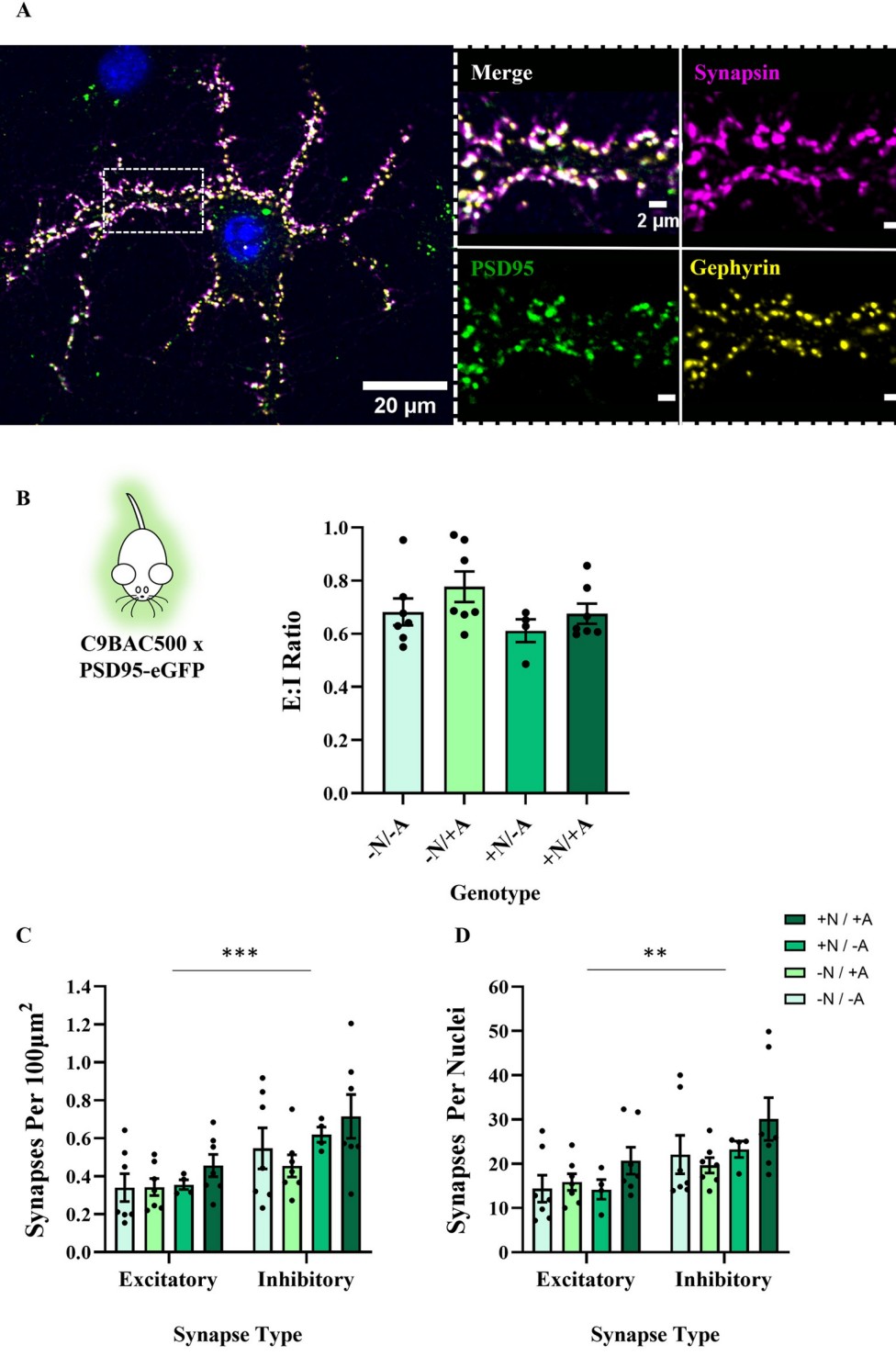

**Fig 2. The presence of the C9BAC500 expansion in astrocytes or neurons has no impact on E:I ratios in primary co-cultures. A**) Example image of DIV 21 neuron in a C9BAC500 x PSD95-eGFP co-culture expressing synapsin, PSD95-eGFP and gephyrin. When zoomed, we see clear co-localisation of presynaptic and postsynaptic markers, indicating significant synapse formation. **B**) Bar chart showing E:I ratios in co-cultures of 4 different genotype combinations. + /—refers to the C9BAC500 genotype and N / A refers to the cell type, either neurons or astrocytes, respectively (N = 4–7 co-culture platedowns). **C**) Bar chart showing numbers of excitatory and inhibitory synapses normalised to area (N = 4–7 co-culture platedowns). **D**) Bar chart showing numbers of excitatory and inhibitory synapses normalised to nuclei count (N = 4–7 co-culture platedowns).

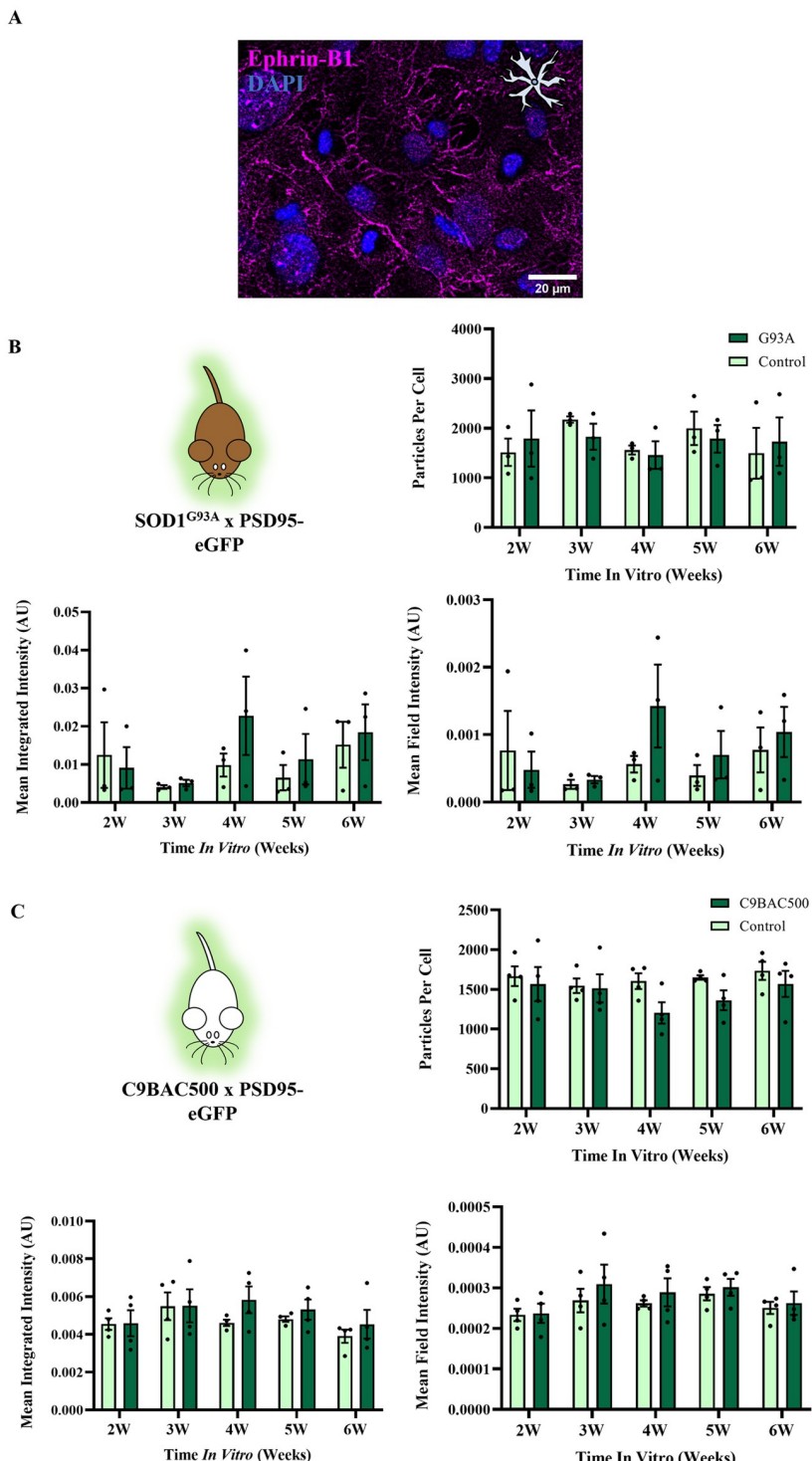

**Fig 3. SOD1^(G93A) and C9BAC500 astrocytes do not express differing levels of synaptogenic protein ephrin-B1 versus controls. A**) Example image of ephrin-B1 expression in primary astrocyte culture. **B**) Expression measures of ephrin-B1 in primary astrocytes harbouring the SOD1^(G93A) mutation. 'G93A' refers to cultures derived from SOD1^(G93A) ^(+/-) PSD95-eGFP ^(+/-) animals, whilst 'Control' refers to cultures derived from SOD1^(G93A) ^(-/-) PSD95-eGFP ^(+/-) animals (N = 3 platedowns). **C**) Expression measures of ephrin-B1 in primary astrocytes harbouring the C9BAC500 mutation. 'C9BAC500' refers to cultures derived from C9BAC500 ^(+/-) PSD95-eGFP ^(+/-) animals, whilst 'Control' refers to cultures derived from C9BAC500 ^(-/-) PSD95-eGFP ^(+/-) animals (N = 4 platedowns).

behavioural changes; with animals being relatively sessile in week 1, initiating basic weight bearing locomotion in week 2, and then being able to produce complex motor behaviours similar to that of adults during week 3 [41]. We therefore selected week 3 as an appropriate, early presymptomatic timepoint when any biases in E:I ratios could be observed in the relatively 'mature' spinal cord avoiding transient developmental changes. It is also a time point immediately after the peak of postnatal synaptogenesis in the spinal cord, allowing us to look at E:I ratios when this has largely concluded [22].

We obtained perfusion-fixed spinal cord sections from week 3 progeny of SOD1$^{G93A}$ x PSD95-eGFP mice, and immunohistochemistry was performed to label synapsin and gephyrin, as in our co-culture study. Tiled, high-resolution maps were obtained across entire spinal cord hemi-sections in order to quantify excitatory and inhibitory synapses in different regions of the spinal cord (Rexed's laminae: L1/2, 3/4, 5, 6, 7, 8, 9, 10). Lamina 1 is the most dorsal, descending to lamina 8 and 9 which represent the site of the medial and lateral MN pools, respectively (see Fig 4A). Widescale mapping of E:I ratios across spinal laminae revealed no differences between control and SOD1$^{G93A}$ mice ($F(1,48) = 2.423$, $P = 0.13$) (see Fig 4C). Similarly, we observed no differences in the numbers of excitatory or inhibitory synapses between control and SOD1$^{G93A}$ mice (excitatory: $F(1,48) = 0.06282$, $P = 0.80$, inhibitory: ($F(1,48) = 1.270$, $P = 0.27$). We did observe inter-regional differences in E:I ratios ($F(7,48) = 6.551$, $P < 0.0001$) that were accounted for by significant diversity in the number of excitatory synapses between different laminae (as has been reported previously [42]), while overall there was no statistical difference in the number of inhibitory synapses between laminae (see Fig 4D). Therefore, we observe no changes in excitatory or inhibitory synapse number in the young SOD1$^{G93A}$ spinal cord.

## E:I ratios are unaffected in human iPSC-derived MN/astrocyte cultures harbouring C9orf72 repeat expansions

Finally, we asked whether there was evidence of astrocyte-driven changes in synaptic E:I ratios in a human cell model of ALS. IPSC-derived MNs have been demonstrated to recapitulate some of the molecular and cellular mechanisms of ALS, including changes in neuronal excitability, synaptic dysfunction and non-cell autonomous pathology [6, 43]. We therefore wanted to use them to look for evidence of changes in E:I ratios, outside of mouse models. We generated human iPSC-derived cultures comprising MNs, other neuronal cell types and astrocytes to investigate whether cultures derived from ALS patients displayed changes in excitatory or inhibitory synapse numbers. Three replicate batches of iPSC-derived cultures were prepared from an iPSC line harbouring a C9orf72 hexanucleotide repeat expansion along with a gene corrected isogenic control line in which the hexanucleotide repeat sequence had been removed [44].

Synapse numbers were quantified at 28 days post-plating, when functional synaptic activity is reliably reported and early indications of hyperexcitability are observed [6]. Fig 5A demonstrates a representative image of our E:I markers in these iPSC-derived MN / astrocyte cultures. Excitatory synapse formation was abundant, apparent from juxtaposed synapsin and PSD95 puncta. In contrast, inhibitory synapses with juxtaposed synapsin and gephyrin puncta were observed infrequently in our iPSC-derived cultures. In separate analyses, gephyrin puncta were also overlapping with more specific inhibitory presynaptic markers GAD65 and VGAT, indicating that despite their relatively low occurrence in these iPSC-derived co-cultures, these are bona-fide inhibitory synapses (S4 Fig).

As expected, there were considerably more excitatory synapses in these cultures than inhibitory synapses ($F(1,8) = 83.21$, $P < 0.0001$) (see Fig 5B). These findings are consistent with

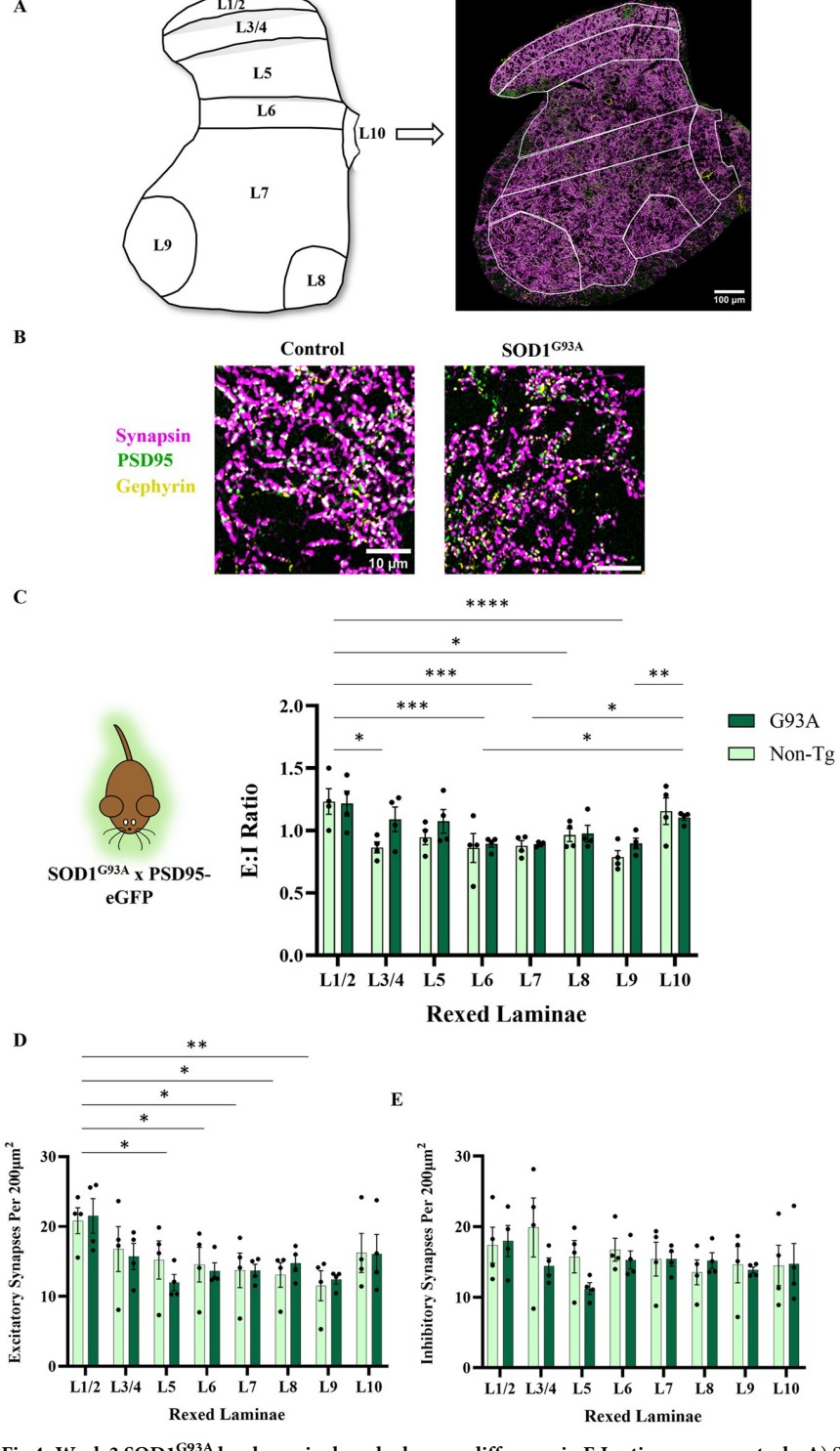

**Fig 4. Week 3 SOD1<sup>G93A</sup> lumbar spinal cords show no difference in E:I ratios versus controls.** **A**) Schematic demonstrating the Rexed's laminae of the L3 lumbar spinal cord used to delineate E:I ratios in different parts of the grey matter, together with a hemi-section scan of synapsin, PSD95-eGFP and gephyrin. **B**) Example zoomed images revealing high resolution synaptic structures of a control spinal cord (SOD1$^{G93A\,-/-}$ PSD95-eGFP $^{+/-}$) and a SOD1$^{G93A}$ spinal cord (SOD1$^{G93A\,+/-}$ PSD95-eGFP $^{+/-}$). **C**) Quantification of E:I ratios across spinal laminae in control versus SOD1$^{G93A}$ spinal slices (P16-P19, N = 4 for control and SOD1$^{G93A}$). **D**) Quantification of excitatory synapse density across spinal laminae in control versus SOD1$^{G93A}$ spinal slices (N = 4 for control and SOD1$^{G93A}$). **E**) Quantification of inhibitory synapse density across spinal laminae in control versus SOD1$^{G93A}$ spinal slices (N = 4 for control and SOD1$^{G93A}$).

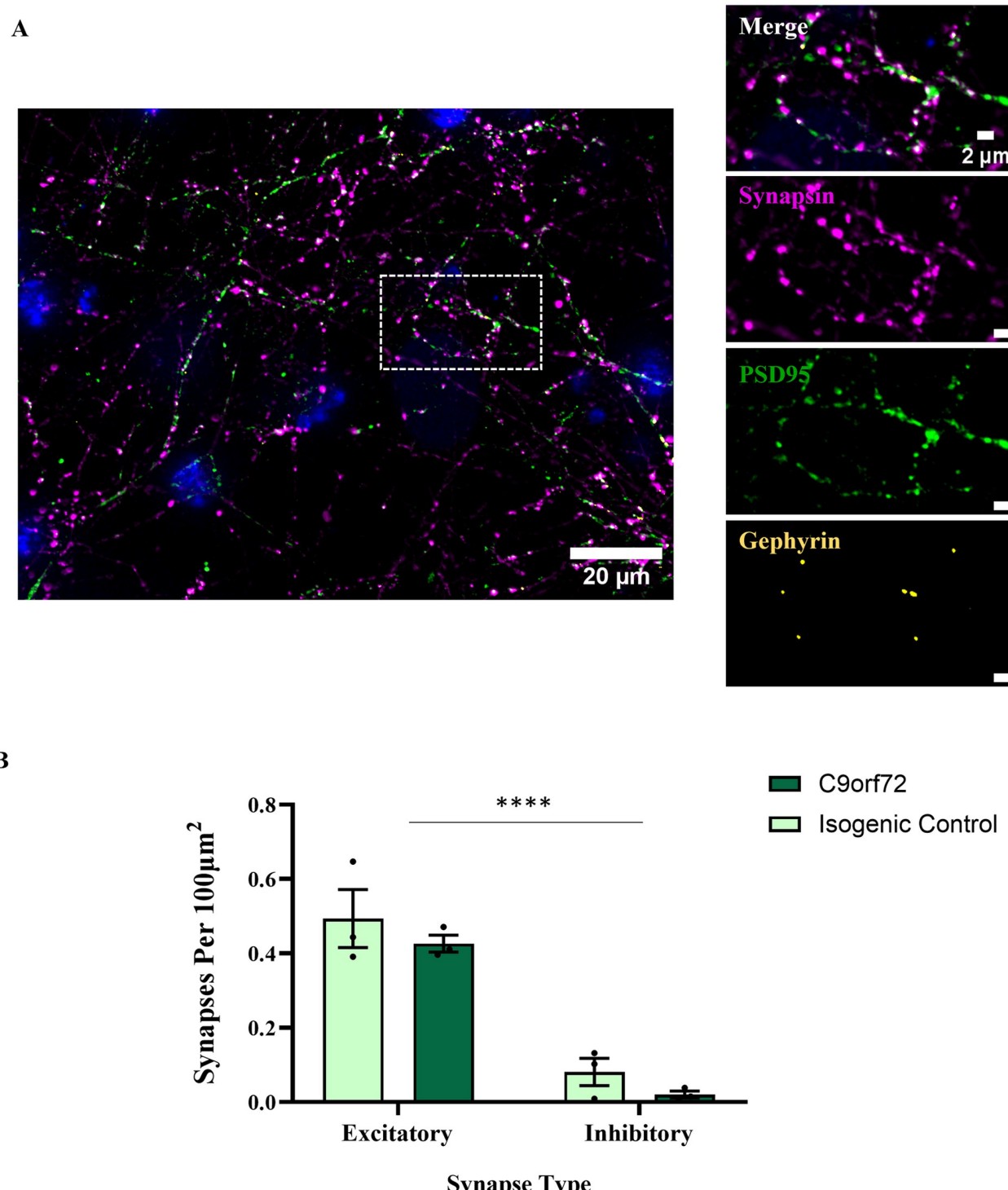

**Fig 5. IPSC-derived MN/astrocyte cultures harbouring a C9orf72 repeat expansion show no changes in synapse formation compared to isogenic controls. A**) Example image of a 28 day post-plating iPSC-derived MN/astrocyte culture expressing synapsin, PSD95 and gephyrin. Zoomed panels on the right show a large degree of synapsin-PSD95 co-localisation, with relatively few gephyrin puncta present. **B**) Quantification of excitatory vs inhibitory synapse densities in C9orf72 cultures vs isogenic controls (N = 3).

previous reports that the vast majority of synapses formed are excitatory in similar cultures [6, 45]. Neither excitatory ($T(4) = 0.8318$, P = 0.45) nor inhibitory ($T(4) = 1.594$, P = 0.19) synapse densities were different between disease conditions. Importantly, the degree of bias towards excitatory synapse formation was not dependent on the genotype of the cells ($F(1,8) = 0.006170$, P = 0.94), together indicating that the presence of the C9orf72 hexanucleotide repeat did not cause a pathological shift towards excitation.

## Discussion

We hypothesised that network-wide hyperexcitability in ALS could be, in part, caused be an early shift in the E:I ratio towards excitation. Furthermore, we hypothesised that any synaptic phenotype could be driven by cell autonomous (neuronal) and/or non-cell autonomous (astrocytic) mechanisms. From analysis of excitatory and inhibitory synapse numbers in combinatorial co-cultures of diseased and healthy neurons and astrocytes we observed no such synaptic phenotype. This finding was confirmed using two different ALS mouse models, SOD1$^{G93A}$ and C9BAC500.

Our study focussed on mouse tissue and cell culture models that would facilitate investigation of the earliest indications of synaptic imbalance. Previous work in our laboratory has established that SOD1$^{G93A}$ animals in our colony first exhibit motor symptoms (hind-limb splay and tremor) at approximately 76 days of age [46], which is consistent with previous reports [47]. We have previously reported widescale synaptic loss and structural changes in spinal cord synapses in SOD1$^{G93A}$ mice aged P90-120 [46, 48], though others have reported synaptic changes in SOD1$^{G93A}$ mice at a range of early timepoints from 14–40 days [49–51]. C9BAC500 animals have previously been shown to display a complex phenotype, with some gait abnormalities present by approximately 16 weeks (112 days), and 30–35% of females developing a rapidly progressive disease, characterised by hindlimb weakness and paralysis, between 20–40 weeks [33]. However, whilst we did observe G4C2 RNA foci in our own previous investigation of C9BAC500 mice, we did not observe any synaptic changes or overt motor dysfunction up to 22 weeks (154 days) [46]. It should be noted that variability in the pathological penetrance of this mouse model has been highlighted by others [52, 53]. Confirmation of our findings using two different ALS mouse models is important, as although the C9BAC500 model has been challenged for its incomplete phenotypic penetrance [52, 53], the lack of effect in the robustly phenotypic SOD1$^{G93A}$ model provides greater confidence in our observations.

The lack of any ALS-related E:I shift was paralleled by a lack of change in the expression of ephrin-B1, a known contact-dependent synaptogenic factor which controls E:I ratios. In the hippocampus during a major period of synaptogenesis, ephrin-B1 can act to modulate E:I ratios. Downregulation of astrocytic ephrin-B1 causes an increase in excitatory synapse formation as well as loss of inhibitory synapses, which appears to at least partially be due to parvalbumin +ve inhibitory neuron loss [24, 40]. A lack of disease-related change in ephrin-B1 expression provides further confidence in the conclusion that there are no ALS-related E:I shifts in our primary co-culture model, either through astrocytic or neuronal mechanisms.

In order to see if intact spinal circuitry shows alterations in E:I ratios in a mouse model of ALS, we visualised synapses across grey matter laminae from the lumbar spinal cords of week 3 control and SOD1$^{G93A}$ mice, using the same excitatory and inhibitory synaptic markers as our *in vitro* study. We chose to compare SOD1$^{G93A}$ and control animals at week 3, immediately after major periods of synaptogenesis and maturation in order to assess if this early system may be developmentally primed towards excess excitation. To our knowledge, this is the first instance of a complete synaptic map encompassing excitatory and inhibitory transmission across upper lumbar laminae in an ALS mouse model at this developmental stage. Irrespective

of genotype, we noted a bias towards increased E:I ratio as a result of increased excitatory synapse density in more dorsal regions. Such a difference in density between the dorsal and ventral horns has been observed previously [42, 46], however, later in development. It may be the case that during such a period of heightened synaptogenesis [21, 22], transient alterations in regional synaptic subtype constitution develop, which may be altered later as a result of synaptic pruning. Presence of the SOD1[G93A] mutation, however, had no effect on E:I ratios, consistent with our *in vitro* dataset. Previous evidence of synaptic subtype loss throughout the grey matter is sparse, with most evidence measuring synapses directly onto MNs, particularly observing inhibitory synapse loss later in disease progression [14, 15, 17, 18, 54]. One study [55] noted GABAergic presynapse loss at a symptomatic stage, and GABAergic / glycinergic loss at end stage in a low-copy number SOD1[G93A] mouse model, quantified using immunohistochemical labelling intensity throughout the grey matter. Comparisons with the timepoints mentioned in this study are challenging due to vastly different disease progressions in the low copy number SOD1[G93A] versus the standard overexpression SOD1[G93A] model, but it appears to suggest that inhibitory loss is a later-stage phenomenon possibly reflecting interneuron cell death, and not the developmental excitatory shift we hypothesised. We also observed no evidence of E:I shifts across the spinal cord, including lamina 8 and 9, which represent medial and lateral motor pools (and surrounding cell populations), respectively. We consider that any changes in network excitability in the SOD1[G93A] mouse model at early stages of development are not due to overall changes in the number of excitatory or inhibitory synapses. Such changes in synapse number likely occur at a later stage in disease development. Other changes to synapses may occur which are not evident in our anatomical dataset, however. It may be the case that the structure of dendritic spines, which has a direct impact on their functional properties [56–58], is altered without a change in our observed PSD95-eGFP signal. Further to this, ultrastructural organisation of the PSD into nanodomains, which again has a direct impact on synaptic transmission [59, 60], could differ at ALS synapses and would not be identifiable using the diffraction limited microscopy technique utilised in this study.

Finally, in an iPSC-derived MN model containing astrocytes, we found no change in the density of excitatory synapses relative to inhibitory synapses as a result of a C9orf72 hexanucleotide repeat. Thus, we find no evidence of early non-cell autonomous shifts in synaptic E:I ratios across multiple ALS models. Human iPSC technology provides a powerful tool with which to probe many ALS disease mechanisms, including synaptic pathology and non-cell autonomous mechanisms in human cells, and, in the process, hopefully aid translatability of findings [6, 43, 61]. While the vast majority of synapses detected were excitatory as previously reported [6, 45], making this model less ideal for assessing E:I changes, it is still informative due to the ability to assess changes in glutamatergic synapse formation in human cells. When excitatory and inhibitory synapse densities were quantified, no pathological changes as a result of the C9orf72 repeat expansion were observed. After 3–4 weeks in culture, iPSC-derived MNs harbouring C9orf72 repeat expansions have been shown to display hyperexcitability, as indicated by increased firing gain, which may result from a reduction in voltage-activated K+ currents [6, 45]. This early hyperexcitability phenotype is later (>6 weeks post-plating) followed by hypoexcitability, characterised by a reduced spiking probability [6]. The later-stage hypoexcitability phenotype is thought to be driven, in part, by non-cell autonomous mechanisms [43]. Reductions in excitatory synapses have also been observed in C9orf72 patient iPSC-derived MNs when maintained for longer in culture (DIV70) [62]. It may be the case, therefore, that such a reduction in synapses parallels losses observed at later stages in animal models, and the lack of E:I changes observed in our study demonstrates an absence of early developmental synaptic alterations. Although our data is informative alongside results gathered from other models, future work expanding this study

with further patient lines (including those harbouring other ALS mutations) would be beneficial.

An important consideration is that we cannot discern the functionality of these synapses based on merely quantifying synapse numbers alone. For example, Fragile X syndrome displays an increase in synapse number but not all are functional or mature [63]. Alternatively, as hypothesised by Kiernan and colleagues [19], a developmental shift towards hyperexcitability could be due to a delay in the GABAergic switch from excitatory to inhibitory transmission. This switch is a result of changes in KCC2 and NKCC1 expression during the early postnatal period, with alterations in the balance of these transporters modulating intracellular Cl$^-$ concentrations [64]. Initially high intracellular chloride concentrations reduce as neurons mature, resulting in a switch from GABAergic transmission causing depolarisation to hyperpolarisation [19, 65]. Indeed, factors such as neonatal stress have been shown to downregulate KCC2 expression during development and cause a resultant delay in the GABAergic switch [64]. It is conceivable therefore that a downregulation of KCC2 expression in ALS could prime the system towards hyperexcitability *without* showing resultant changes in structural markers. Mòdol and colleagues [66] observed no evidence of changes in KCC2 expression in the ventral lumbar spinal cord of SOD1$^{G93A}$ mice at 8, 12 and 16 weeks of age. However, other authors have noted downregulation of KCC2 mRNA transcripts at the symptomatic age of P120 in lumbar MNs, as well as presymptomatically (P80) in hypoglossal MNs [67]. This was reflected in the qualitative assessment of KCC2 protein immunoreactivity in the ventral horn. Therefore, there is tentative evidence of possible KCC2-driven alterations in spinal network excitability, although a more thorough examination of the balance between KCC2 and NKCC1 at very early stages of the disease is necessary to ascertain if this is the case.

To conclude, there does not appear to be structural evidence of developmental shifts in E:I ratios in ALS spinal networks. It is possible that we have failed to capture even earlier changes. However, our evidence from young spinal tissue after the major period of postnatal synaptogenesis would suggest that if this is the case, it is transient and likely does not represent a major driver of later hyperexcitability.

## Supporting information

**S1 Fig. Quantification of cells expressing astrocyte markers GFAP and GS. A**) Representative image demonstrating GFAP and glutamine synthetase (GS) in enriched astrocyte cultures. **B**) Quantification of % cells positive for astrocyte marker GFAP at 2–6 weeks *in vitro*. **C**) Quantification of % cells positive for astrocyte marker GS at 2–6 weeks *in vitro*.
(TIF)

**S2 Fig. Characterisation of postnatal spinal neural cultures. A**) Example whole-cell patch-clamp recording demonstrating spontaneous synaptic activity measured in voltage-clamp mode in spinal neuron cultures at DIV 18. **B**) Pie chart demonstrating the proportion of cells receiving any spontaneous synaptic input versus no clear synaptic input in a 120s gap-free recording in voltage-clamp mode. **C**) Examples of PSD95 expression at 2, 3 and 4 weeks *in vitro*.
(TIF)

**S3 Fig. Validation of PSD95-eGFP puncta. A**) Schematic demonstrating the structure of PSD95-eGFP. **B**) Validation images in C9BAC500$^{+/-}$ PSD95-eGFP$^{+/-}$ neuron cultures stained with an anti-PSD95 antibody (PSD95 AB). White arrows indicate clear co-localisation between GFP and PSD95 AB.
(TIF)

**S4 Fig. Validation of inhibitory synapse formation in iPSC cultures.** Schematic demonstrating expected location of GAD65, VGAT and gephyrin expression, accompanying an example image of co-localised presynaptic inhibitory markers GAD65 and VGAT, overlapping with postsynaptic marker gephyrin.
(TIF)

## Acknowledgments

We would like to thank Dr. Simon Sharples (University of St Andrews) for advice and support regarding our whole-cell patch-clamp validation. We would also like to thank the animal care staff for their hard work and kindness, particularly during the challenge of working through the COVID-19 pandemic. Finally, we would like to thank the members of the Chandran lab for provision of utilised iPSC lines.

## Author Contributions

**Conceptualization:** Calum Bonthron, Sarah Burley, Matthew J. Broadhead, Gareth B. Miles.

**Data curation:** Calum Bonthron.

**Formal analysis:** Calum Bonthron.

**Funding acquisition:** Matthew J. Broadhead, Seth G. N. Grant, Siddharthan Chandran, Gareth B. Miles.

**Investigation:** Calum Bonthron.

**Methodology:** Calum Bonthron, Sarah Burley, Matthew J. Broadhead, Vanya Metodieva, Seth G. N. Grant, Siddharthan Chandran.

**Supervision:** Sarah Burley, Matthew J. Broadhead, Gareth B. Miles.

**Validation:** Calum Bonthron.

**Visualization:** Calum Bonthron.

**Writing – original draft:** Calum Bonthron, Matthew J. Broadhead, Gareth B. Miles.

**Writing – review & editing:** Calum Bonthron, Matthew J. Broadhead, Gareth B. Miles.

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
