## [Decision Letter · Decision Letter 0]

16 May 2024

PONE-D-24-06359Excitatory to inhibitory synaptic ratios are unchanged at presymptomatic stages in multiple models of ALS

PLOS ONE

Dear Dr. Miles,

Thank you for submitting your manuscript to PLOS ONE. After careful consideration, we feel that it has merit but does not fully meet PLOS ONE’s publication criteria as it currently stands. Therefore, we invite you to submit a revised version of the manuscript that addresses the points raised during the review process.

We look forward to receiving your revised manuscript.

Kind regards,

Stephen D. Ginsberg, Ph.D.

Section Editor

PLOS ONE

Journal Requirements:

2. Thank you for stating the following in your Competing Interests section: "None."

**Additional Editor Comments:**

After careful consideration by 2 Reviewers and an Academic Editor, all of the critiques of both Reviewers must be addressed in detail in a revision to determine publication status. If you are prepared to undertake the work required, I would be pleased to reconsider my decision, but revision of the original submission without directly addressing the critiques of the Reviewers does not guarantee acceptance for publication in PLOS ONE. If the authors do not feel that the queries can be addressed, please consider submitting to another publication medium. A revised submission will be sent out for re-review. The authors are urged to have the manuscript given a hard copyedit for syntax and grammar.

**Comments to the Author**

1. Is the manuscript technically sound, and do the data support the conclusions?

Reviewer #1: Yes

Reviewer #2: Yes

2. Has the statistical analysis been performed appropriately and rigorously? 

Reviewer #1: Yes

Reviewer #2: I Don't Know

3. Have the authors made all data underlying the findings in their manuscript fully available?

Reviewer #1: Yes

Reviewer #2: Yes

4. Is the manuscript presented in an intelligible fashion and written in standard English?

Reviewer #1: Yes

Reviewer #2: No

5. Review Comments to the Author

Reviewer #1: This paper details an analysis of excitatory vs inhibitory synapses in the spinal cord, primary co-cultures and iPSC for SOD1G93A and C9orf72 expansion models.

Overall they have not found any differences. They acknowledge that there are things that they have not tested regarding functionality but it is clear there are no overt differences in these models with the presence of an ALS related mutation.

In general, given the negative message in this paper i have no problem with the results that have been presented, although their significance is hard given how unclear the data is around this as a definitive cause of ALS in fully symptomatic models. That said it is important that this is published given the large amount of work that is involved in producing such data so that the field can see that there are no differences.

The only thing that I would like to see would be the inclusion of some details of the phenotypes of the mice that they are using. Whilst i am aware most of these are published models, it would be better to have the details in this paper so that people can see how early these are being studied before symptoms. Again for the iPSC, give some phenotype details so that it is clear what is being actually looked at here in terms of time frames etc. Given these are validated models, do these iPSC show phenotypes that should be taken into account.

Reviewer #2: This is an interesting manuscript analyzing the balance of excitation and inhibition at presymptomatic stages in models of ALS. The manuscript presents some important data, however major revisions are needed.

It seems surprising that the E:I ratio is shown first, followed by the counts of the different synapses. It is expected that the synapses have already been counted, in order to get the ratio..? Why not show these data first, and then the ratio as the last part? In fact, it should be clarified in details how exactly the ratio was calculated, as that is not clear in the present version.

Page 12 line 274 – 279 it is written: “For brevity, +N and +A refer to neurons and astrocytes, respectively, which are positive for the SOD1G93A or C9BAC500 mutation. -N and -A refers to these cell types generated from negative littermates that do not express ALS mutations. This produced 4 conditions in which to investigate non-cell autonomous effects on E:I ratios (+N/+A, +N/-A, -N/+A, -N/-A) (see Fig 1B).”

Based on this explanation, it would be most suitable to display the data derived from -N/-A conditions first in the figures, since that would be close to normal situation, with no mutations expressed. Usually, when preparing figures, the wild-type or control situation is displayed first, followed by the different interventions of increasing complexity. This comment is relevant for all figures.

Similarly, Figure 3 also shows the mutant first followed by the control. This should also be the other way around.

Figure legends could be improved by adding a short sentence in the beginning, as sort of a header. This would help the reader get the conclusion from each figure.

Page 5, line 123. Please specify the specific “C57bl/6” background, as there is more than one strain of these mice.

Page 13 line 299 refers to “Figure 1D”, but there is no panel 1D. Similar in the Figure legend above on page 12. Again, this comment is also relevant for Figure 2. Please correct.

6. PLOS authors have the option to publish the peer review history of their article (what does this mean?). If published, this will include your full peer review and any attached files.

**Do you want your identity to be public for this peer review?** For information about this choice, including consent withdrawal, please see our Privacy Policy.

Reviewer #1: No

Reviewer #2: No

---

## [Author Response · Author response to Decision Letter 0]

5 Jun 2024

Response to Reviewer

We would like to thank both reviewers for their careful consideration of the manuscript, and for acknowledging the importance of our findings for the field. We have now amended the manuscript in light of the helpful reviewer comments. We believe the changes made have improved the manuscript and hope it will satisfy the requirements for publication. In our responses below, and when referring to lines or sections, we refer to changes in the ‘tracked changes’ version of the manuscript.

Reviewer 1:

“The only thing that I would like to see would be the inclusion of some details of the phenotypes of the mice that they are using. Whilst I am aware most of these are published models, it would be better to have the details in this paper so that people can see how early these are being studied before symptoms.”

• We agree that providing a more thorough background on the phenotypes of the mouse models is highly valuable and aids in the interpretation and understanding of our findings. We have therefore provided more substantial information regarding the phenotypes of the mice in our revised Discussion section [431-444].

“Again for the iPSC, give some phenotype details so that it is clear what is being actually looked at here in terms of time frames etc. Given these are validated models, do these iPSC show phenotypes that should be taken into account.”

• Similarly, we have provided more extensive discussion of the phenotypes of iPSC-derived MNs [509-515]. Notably we have included mention of a hyperexcitability phenotype we (and others) have previously seen in iPSC-derived MNs at the timepoint used in our current study. This provides further support for our conclusion that any observed hyperexcitable phenotype is unlikely to be driven by biases in excitatory versus inhibitory synapse formation [396].

Reviewer 2:

“It seems surprising that the E:I ratio is shown first, followed by the counts of the different synapses. It is expected that the synapses have already been counted, in order to get the ratio..? Why not show these data first, and then the ratio as the last part? In fact, it should be clarified in details how exactly the ratio was calculated, as that is not clear in the present version.”

• We quantified the number of excitatory and inhibitory synapses in each individual image, then an E:I value was produced from these values (no. excitatory synapses / no. inhibitory synapses). The ratios for each image were then averaged across mouse or cell line. A more detailed explanation of E:I quantification has been added in the revised Results section [287-289]. 

• We appreciate that these data could be presented in either order (i.e. raw synapse numbers followed by ratios or vice versa). We wanted to present the ratio data first because we think it provides the clearest and simplest summary of the data, and speaks most directly to the main question we set in our project; whether the balance between excitatory and inhibitory synapses is altered at early stages of disease. Subsequently, we show the number of excitatory and inhibitory synapses (normalised to either the area or the number of DAPI nuclei) to verify that the static E:I ratios observed were not masking a scaled increase / decrease of excitatory and inhibitory synapses. We have added a justification for why we present the ratios followed by synapse densities in our revised Results section [285-287].

“Based on this explanation, it would be most suitable to display the data derived from -N/-A conditions first in the figures, since that would be close to normal situation, with no mutations expressed. Usually, when preparing figures, the wild-type or control situation is displayed first, followed by the different interventions of increasing complexity. This comment is relevant for all figures.”

• All relevant figures (Fig. 1, 2, 3, 4, 5 and Supp. Fig. 1) have been amended accordingly.

“Figure legends could be improved by adding a short sentence in the beginning, as sort of a header. This would help the reader get the conclusion from each figure.”

• All figure and supplementary figure legends now include an initial title sentence.

“Page 5, line 123. Please specify the specific “C57bl/6” background, as there is more than one strain of these mice.”

• We have now included full details of the mouse lines [124-128].

“Page 13 line 299 refers to “Figure 1D”, but there is no panel 1D. Similar in the Figure legend above on page 12. Again, this comment is also relevant for Figure 2. Please correct.”

• We thank the reviewer for noticing this error. Panel denotation for Figure 1D and 2D have been added to the figures.

---

## [Decision Letter · Decision Letter 1]

18 Jun 2024

Excitatory to inhibitory synaptic ratios are unchanged at presymptomatic stages in multiple models of ALS

PONE-D-24-06359R1

Dear Dr. Miles,

We’re pleased to inform you that your manuscript has been judged scientifically suitable for publication and will be formally accepted for publication once it meets all outstanding technical requirements.

Kind regards,

Stephen D. Ginsberg, Ph.D.

Section Editor

PLOS ONE

**Comments to the Author**

1. If the authors have adequately addressed your comments raised in a previous round of review and you feel that this manuscript is now acceptable for publication, you may indicate that here to bypass the “Comments to the Author” section, enter your conflict of interest statement in the “Confidential to Editor” section, and submit your "Accept" recommendation.

Reviewer #1: All comments have been addressed

Reviewer #2: All comments have been addressed

2. Is the manuscript technically sound, and do the data support the conclusions?

Reviewer #1: Yes

Reviewer #2: Yes

3. Has the statistical analysis been performed appropriately and rigorously? 

Reviewer #1: Yes

Reviewer #2: Yes

4. Have the authors made all data underlying the findings in their manuscript fully available?

Reviewer #1: Yes

Reviewer #2: Yes

5. Is the manuscript presented in an intelligible fashion and written in standard English?

Reviewer #1: Yes

Reviewer #2: Yes

6. Review Comments to the Author

Reviewer #1: The authors have addressed all the comments and i am happy to proceed this to publication. Well Done!

Reviewer #2: The comments have satisfactorily been addressed. The manuscript has been significantly improved by the revision. The reviewer has no further comments.

7. PLOS authors have the option to publish the peer review history of their article (what does this mean?). If published, this will include your full peer review and any attached files.

Reviewer #1: No

Reviewer #2: No

---

## [Editor Report · Acceptance letter]

28 Jun 2024

PONE-D-24-06359R1 

PLOS ONE

Dear Dr. Miles, 

I'm pleased to inform you that your manuscript has been deemed suitable for publication in PLOS ONE. Congratulations! Your manuscript is now being handed over to our production team.

Kind regards, 

on behalf of

Dr. Stephen D. Ginsberg 

Section Editor

PLOS ONE